# 3-Bromopyruvate Impairs Mitochondrial Function in *Trypanosoma cruzi*

**DOI:** 10.3390/pathogens14070631

**Published:** 2025-06-25

**Authors:** Rafaella Oliveira da Costa, Davi Barreto-Campos, Juliana Barbosa-de-Barros, Giovanna Frechiani, Luiz Fernando Carvalho-Kelly, Ayra Diandra Carvalho-de-Araújo, José Roberto Meyer-Fernandes, Claudia Fernanda Dick

**Affiliations:** 1Institute of Medical Biochemistry Leopoldo de Meis, Federal University of Rio de Janeiro, Rio de Janeiro 21590-902, RJ, Brazil; rafaellaodacosta@gmail.com (R.O.d.C.); davi_barreto@biof.ufrj.br (D.B.-C.); ju.bb998@biof.ufrj.br (J.B.-d.-B.); lfkelly@bioqmed.ufrj.br (L.F.C.-K.); ayra.araujo@bioqmed.ufrj.br (A.D.C.-d.-A.); 2Graduate Program in Translational Biomedicine/BIOTRANS, Grande Rio University, Duque de Caxias 25071-202, RJ, Brazil; giovanna-degering@biof.ufrj.br; 3Program of Structural Biology and Bioimaging, Center for Health Sciences, Federal University of Rio de Janeiro, Rio de Janeiro 21941-902, RJ, Brazil

**Keywords:** Chagas disease, energy metabolism, 3-bromopyruvate

## Abstract

*Trypanosoma cruzi* is a kinetoplastid parasite and etiological agent of Chagas disease. Given the significant morbidity and mortality rates of this parasitic disease, possible treatment alternatives need to be studied. 3-Bromopyruvate (3-BrPA) is a synthetic analog of pyruvate that was introduced in the early 21st century as an anticancer agent, affecting the proliferation and motility of certain microorganisms. Therefore, this work aims to evaluate the role of 3-BrPA in the energy metabolism, proliferation, and infectivity of *T. cruzi*, with a primary focus on the mitochondrial state, ATP production, and the key glycolytic pathway enzymes. It was observed that mitochondrial function in 3-BrPA cells was impaired compared to control cells. Accordingly, cells maintained in control conditions have a higher intracellular ATP content than cells maintained with 3-BrPA and higher ecto-phosphatase activity. However, the 3-BrPA reduced ecto-nuclease activity and was capable of hydrolyzing 5′-AMP, ADP, and ATP. When we evaluated two key glycolytic pathway enzymes, glucose kinase (GK) and glyceraldehyde-3-phosphate dehydrogenase (GAPDH), we observed that 3-BrPA induced higher GAPDH activity but did not alter GK activity. The compensatory energy mechanisms presented in *T. cruzi* may influence the process of cell metabolism and, consequently, the functional infectious process, suggesting the potential use of 3-BrPA in future clinical applications for Chagas disease.

## 1. Introduction

Chagas disease (CD), named in recognition of its discoverer [1], is a parasitic disease caused by the protozoan *Trypanosoma cruzi*. *T. cruzi* is a kinetoplastid parasite with a complex life cycle, alternating between a vertebrate host (mammals, including humans) and an invertebrate host (insect vector). Nowadays, the most frequent transmission mode is through ingesting sugarcane juice or food contaminated with infected Triatominae or the feces of this insect vector [2]. In Brazil, approximately 76% of CD cases were transmitted orally in 2021, while 7% were transmitted by the vector [3]. However, transplacental contamination from an infected mother to her fetus is also possible [4]. Less than 20% of cases occur through blood transfusion and organ transplantation from an infected donor, since centers have strict control to detect the infection before transfusion [5]. The classic form of transmission of the parasite to humans is through the feces of hematophagous triatomines (such as *Triatoma infestans* and *Rhodnius prolixus*) during insect bites, through fission in the skin or mucous membranes (such as the conjunctiva, causing the so-called Romaña sign; cf. [6]), or by oral/digestive means, in this case through contaminated food [7,8]. Chagas disease is a neglected disease, as it mainly affects low-income populations with restricted access to health services due to their location with little professional care, especially in remote rural areas and slums, and due to a lack of resources. However, this is an endemic parasitic disease in 21 Latin American countries, and there are cases in 44 countries worldwide, with 75 million people at risk of infection [2].

3-Bromopyruvate (3-BrPA) is an analog of pyruvate, an alkylating agent that can regulate enzyme activity [9] and react with the thiol and hydroxyl groups of enzymes [10]. The evaluation of 3-BrPA has become promising because this compound targets metabolic pathways. This compound was initially described as an antitumor agent that induces cell death in many cancer cells in vivo [11]. After that, it is established as an inhibitor of hexokinase II by promoting modifications of cysteine residues [12,13], an inhibitor of the enzyme glyceraldehyde-3-phosphate dehydrogenase in the glycolytic pathway [14,15,16], and succinate dehydrogenase in the endoplasmic reticulum [16]. Protozoa and cancer cells, highly proliferative cell models, rely on glycolytic pathways for ATP production. This rapid replication is involved in metabolic processes that efficiently convert glucose and specific amino acids into biomass and energy at expected rates [17]. In this way, 3-BrPA soon proved effective against microorganisms, inhibiting the proliferation of *Saccharomyces cerevisiae* [18], *Cryptococcus neoformans* [19], and *Toxoplasma gondii* [20]. In *Trypanosoma brucei*, motility was inhibited, and viability was reduced in cells grown with 3-BrPA [21,22]. 3-BrPA showed a slight parasite-killing effect in *Leishmania infantum* promastigotes. Moreover, it presents an inhibitory dose-dependent effect on *L. infantum* intramacrophage amastigotes. 3-BrPA also demonstrated antiparasitic effects, as it promotes a decrease in *T. brucei* survival and reduces the load of intramacrophage amastigotes of *L. infantum* [23]. In *L. amazonensis* promastigotes, the inhibition of enzymes in the glycolytic pathway (glucose kinase, glyceraldehyde-3-phosphate dehydrogenase, and enolase kinase) has been described, along with a reduction in ATP production and O_2_ consumption, which reduces by 50% the capacity of these cells to infect macrophages when grown with 3-BrPA [24]. Although 3-BrPA shows no effects on *T. cruzi* intracellular amastigote form proliferation [23], possible impacts on parasite metabolism were not evaluated. In this way, the present study aimed to evaluate the effects of 3-BrPA on the energy metabolism of epimastigote forms of *T. cruzi*, especially on highly proliferative epimastigote forms, thereby promoting the reuse of a compound that has already proven effective in cancer treatment.

## 2. Materials and Methods

### 2.1. Cell Culture and Proliferation Profile

Epimastigote forms were maintained in LIT medium (“Liver Infusion Tryptose”) at 28 °C, adjusted to pH 7.2 with HCl, and supplemented with 10% fetal bovine serum (Cripion, São Paulo, Brazil), with or without the addition of 300 µM 3-Bromopyruvate. There was no effect of 300 µM 3-BrPA on systemic toxicity [25]. The cell density was estimated using a hemocytometer chamber, and the proliferation curve was monitored starting with the same cell inoculum. The cell proliferation was monitored daily.

### 2.2. Cell Viability

Cell viability was checked before counting by observing parasite mobility and using Trypan blue exclusion dye [24]. The MTT assay also assessed cell viability [26]. Disrupted freeze–thaw cells were used as a negative control.

### 2.3. Hypoxic Induction

A total of 5 × 10^7^ cells/well were placed into the hypoxia chamber to induce hypoxia [27]. CO_2_ was absorbed with breath chalk, and the O_2_ partial pressure (pO_2_) was monitored continuously. The isobaric N_2_O/O_2_ mixture was injected with oxygen until it contained 20.9 vol.% oxygen (equivalent to atmospheric oxygen levels) within 7 min. The pO_2_ level decreased to 5 vol.% and maintained for 10 min, after which the cells were incubated for an additional 60 min with or without 3-BrPA treatment.

### 2.4. Inorganic Phosphate Transport

The transport of Pi was evaluated as described earlier [28]. Intact epimastigotes (1.0 × 10^7^ cells/mL) were maintained at 25 °C for 60 min in a reaction mixture consisting of 140 mM NaCl or 140 mM choline chloride, 1.5 mM CaCl_2_, 5 mM KCl, 10 mM HEPES–Tris (pH 7.4), 1 mM MgCl_2_, and 100 μM ^32^Pi (2.5 mCi/μmol). The addition of ^32^Pi triggered the reaction, which was stopped by the addition of 0.2 mL of a cold solution containing 140 mM choline chloride, 1.5 mM CaCl_2_, 5 mM KCl, 10 mM HEPES–Tris (pH 7.4), and 1 mM MgCl_2_. After three washes with the same cold buffer, the cells were lysed by adding 0.1% SDS, and the radioactivity was quantified using a scintillation counter. The cells were treated with a cold ^32^Pi reaction mixture to determine the blank uptake values and then stored on ice for 60 min.

### 2.5. Ecto-Phosphatase Activity

Phosphatase activity was determined by using p-nitrophenylphosphate (pNPP) as the substrate and by measuring the p-nitrophenol (p-NP) production rate. Intact epimastigote cells (3 × 10^9^ cells) were cultivated with or without 3-BrPA and incubated in a reaction mixture containing 116 mM NaCl, 5.4 mM KCl, 5.5 mM glucose, 50 mM HEPES (pH 7.2), and 0.8 mM MgCl_2_. The reaction was started by adding 5 mM pNPP. After 60 min, the reactions were stopped by adding 1 N NaOH, and the results were determined spectrophotometrically at 425 nm [29]. A p-NP curve was plotted and used as a standard to determine the concentration of released p-NP. The ecto-phosphatase activity was calculated as p-NP released per cell number.

### 2.6. Ecto-5′-Nucleotidase Activity

The ecto-5′-nucleotidase activity was assessed by measuring the rate at which inorganic phosphate (Pi) was released. Intact cells (3 × 10^9^ cells) were incubated for 1 h at room temperature in a reaction mixture that contained 116 mM NaCl, 5.4 mM glucose, 50 mM HEPES–Tris buffer (pH 7.4), and 5 mM 5′AMP as the substrate. The reaction was stopped by adding 1 mL of 25% charcoal in 0.1 M HCl. Following the reaction, the tubes were centrifuged at 1500× *g* for 15 min at 4 °C, and 0.1 mL of the supernatant was combined with 0.1 mL of the Fiske–Subbarow reactive mixture [30]. The released Pi’s absorbance was determined using a spectrophotometer at a wavelength of 650 nm. The ecto-5’-nucleotidase activity was calculated using a standard Pi curve and adjusted per cell number.

### 2.7. Glucokinase Activity

First, *T. cruzi* epimastigotes were grown for 6 days, with or without 3-BrPA. Then, they were harvested and resuspended in a lysis buffer containing 10 mM Tris/HCl (pH 7.0), 20 mM NaF, 1 mM dithiothreitol (DTT), 250 mM sucrose, 5 mM EDTA, 1 mM PMSF, 10 μM leupeptin, 1 μM pepstatin A, and 0.1% Triton X-100, following three freezing and thawing cycles in liquid nitrogen to disrupt the cells [24]. The protein concentration was determined as described before [31]. Then, cellular extracts were incubated in a reaction buffer containing 20 mM Tris: HCl (pH 7.4), 5 mM MgCl_2_, 1 mM glucose, 1 unit/mL glucose-6-phosphate dehydrogenase (G6PDH) (*Leuconostoc mesenteroides*), 0.1% Triton X-100, 5 mM NaN_3_, 1 mM ATP, and 50–100 μg/mL protein. After a 3 min pre-incubation, the reaction was initiated by adding 0.5 mM β-NADP^+^ and was quantified spectrophotometrically by the reduction of β-NADP^+^ to β-NADPH at 340 nm. Total NADPH generation was determined using the NADPH standard curve.

### 2.8. Total Glyceraldehyde 3-Phosphate Dehydrogenase Activity

The total GAPDH activity was assessed as the conversion of NAD^+^ to NADH [32]. Cellular extracts were incubated in a reaction buffer containing 100 mM triethanolamine-HCl buffer (pH 7.5) with 5 mM MgSO_4_, 1 mM EDTA, 1 mM DTT, 30 mM KH_2_AsO_4_, and 1.5 mM NAD^+^. The reaction was initiated by adding 2 mM glyceraldehyde 3-phosphate and incubating for 15 min at 37 °C. NADH generation was quantified spectrophotometrically at 340 nm every minute for 5 min. Total NADH generation was determined using the NADH standard curve.

### 2.9. Intracellular ATP Measurement

Intracellular ATP (ATPi) levels were assessed using an ATP bioluminescent assay kit (Sigma-Aldrich, St. Louis, MO, USA). Briefly, epimastigotes (1 × 10^7^ parasites per tube) were incubated in a solution consisting of 100 mM sucrose, 50 mM KCl, and 50 mM Tris-HCl (pH 7.2). The parasites were mixed with a somatic cell ATP-releasing reagent to prepare cellular extracts, and then the combination was chilled for 1 min. The resulting mixture was transferred to MTS-11C mini tubes that contained the ATP assay mix (Axygen, Union City, CA, USA) and swirled for 10 s at room temperature. A GloMax Multi JR detection system from Promega (Madison, WI, USA, EUA) was used to quantify the total light emitted. In each experiment, the intracellular ATP concentration was calculated using a standard ATP curve and adjusted per cell number [27].

### 2.10. High-Resolution Respirometry

The oxygen consumption of intact epimastigotes (5 × 10^7^ parasites/chamber) was measured using an O2k-system high-resolution oxygraph (Oxygraph-2K; Oroboros Instruments, Innsbruck, Austria) at 28 °C with continuous stirring. The cells were suspended in a 2 mL respiration solution containing 100 mM sucrose, 50 mM KCl, and 50 mM Tris–HCl (pH 7.2), and 50 μM digitonin was added to permeabilize the parasites. Oxygen concentrations and O_2_ consumption were recorded using DatLab 7.4 software coupled to Oxygraph-2K (basal O_2_ consumption). Following that, 10 mM succinate and 200 μM ADP were added. Uncoupled respiration was induced with 3 μM carbonyl cyanide 4-(trifluoromethoxy)phenylhydrazone (FCCP) and then blocked with 2.5 μg/mL antimycin A to assess residual O_2_ consumption [27]. In a series of experiments, titration with substrates and inhibitors was not performed, and O_2_ consumption was measured only in the presence of endogenous substrates (basal consumption) throughout all recordings.

### 2.11. Amplex^®^ Red Peroxidase Assay

The production of H_2_O_2_ was assayed by the rate of H_2_O_2_ reduction to H_2_O, which is stoichiometrically coupled (1:1) to the simultaneous oxidation of the non-fluorescent Amplex^®^ Red probe to the fluorescent resorufin. Briefly, 10^7^ parasites/mL, grown with or without 3-BrPA, were incubated in a reaction solution containing 5 mM Tris–HCl (pH 7.4), 1.7 μM Amplex^®^ Red (Invitrogen, Carlsbad, CA, USA), and 6.7 U/mL horseradish peroxidase (Sigma-Aldrich, St. Louis, MO, USA) for 30 min at room temperature. The evolution of fluorescence was monitored at excitation and emission wavelengths of 563 nm and 587 nm, respectively (slit width 5 nm). The H_2_O_2_ production was calculated using a standard H_2_O_2_ curve and adjusted per µg of protein.

### 2.12. Immunofluorescence Microscopy

Protozoa were washed in PBS and incubated with 500 nM MitoTracker^®^ Red CMXRos (Invitrogen, Carlsbad, CA, USA) for 30 min. The cells were washed in PBS and fixed for 1 h with freshly prepared 4% formaldehyde. After fixation, cells were adhered to poly-l-lysine-coated microscope coverslips and permeabilized for 5 min with 1% Triton X-100. Samples were incubated in a blocking solution containing 3% bovine serum albumin (BSA) and 0.02% Tween 20 diluted in PBS, pH 8.0. Next, slides were incubated for 1 h with an antibody produced against aldolase as a glycosome marker [33], diluted in a blocking solution (1:1500). Then, cells were washed with PBS and incubated for 45 min with Alexa488-conjugated anti-rabbit IgG (Molecular Probes, Eugene, OR, USA) diluted 1:500 in blocking solution. Cells were then incubated with 5 μg/mL of DAPI for 30 min. Slides were mounted using the anti-fading reagent ProLong Gold (Invitrogen, Carlsbad, CA, USA) and observed on a ZEISS LSM 910 microscope (ZEISS, Oberkochen, German).

### 2.13. Statistical Analysis

The data are presented as the mean ± standard error of the mean (SEM). An unpaired Student’s *t*-test was conducted to compare the two means. A one-way ANOVA followed by Tukey’s test was utilized when comparing more than two means, as indicated in the text or figure legends. Prior to each ANOVA analysis, the normality of the distribution was evaluated. The significance level of *p* < 0.05 was established. The statistical analysis and figure preparation were performed using GraphPad Prism 7.0 (GraphPad, San Diego, CA, USA).

## 3. Results

As previously demonstrated for amastigotes [22], epimastigote proliferation in vitro is not influenced by the addition of 300 µM 3-BrPA (Figure 1a), with no changes in cell viability, as measured by two different methodologies (Figure 1b,c).

However, the incubation of 3-BrPA changes the activity of proteins and transporters on the cell surface differently. 3-BrPA slightly inhibits the Na^+^-independent Pi transporter (Figure 2a) with no changes on the Na^+^-dependent Pi transporter (Figure 2b). Regarding ecto-enzymes, it was shown that high ecto-phosphatase activity was observed on cells grown in the presence of 3-BrPA (Figure 2c). At the same time, 5’-ectonucleotidase, an important nucleotidase capable of producing adenosine and Pi, is inhibited by 3-BrPA (Figure 2d), indicating a different modulation of *T. cruzi* metabolism.

Higher ecto-phosphatase activity could indicate changes in energy metabolism, as some metabolic enzymes rely on Pi for their function [34]. Basal O_2_ consumption was impaired when cells were incubated with 3-BrPA (Figure 3a). The absence of significant modulation in O_2_ consumption following oligomycin treatment, an FoF1-ATP synthase inhibitor, suggests impaired electron transport chain activity. O_2_ consumption in permeabilized cells (Figure 3b) was analyzed with digitonin to check this. Adding succinate and ADP, substrates for succinate dehydrogenase, did not stimulate O_2_ consumption, reinforcing the impairment of electron transport chains. The addition of FCCP, H^+^ ionophore (electron transfer system uncoupled from phosphorylation), was significantly lower in cells with 3-BrPA. The addition of antimycin A abolishes residual respiration in both cases. The impairment of electron transport could interfere with reactive oxygen production. For that, H_2_O_2_ production was observed (Figure 3d). Accordingly, 3-BrPA induces low H_2_O_2_ production. Total intracellular ATP was measured to confirm the mitochondrial malfunction induced by 3-BrPA (Figure 3d). Intracellular ATP is lower in cells grown with 3-BrPA than in control conditions. This intracellular ATP reduction was observed in hypoxic conditions, regardless of whether 3-BrPA treatment was present. This could indicate that 3-BrPA could induce hypoxia-like conditions.

Once cell proliferation is not changed by treatment with 3-BrPA, regardless of the disruption of mitochondrial function, this could indicate that other metabolic pathways are compensating for the energy loss. Therefore, the glycolytic activities of glucose kinase (Figure 4a) and glyceraldehyde-3-phosphate dehydrogenase (Figure 4b) were evaluated, both of which are potential targets for 3-BrPA inhibition. In this way, it was observed that 3-BrPA can significantly inhibit GAPDH activity to levels comparable to those of iodoacetamide (IA), an irreversible inhibitor of GAPDH. At the same time, 3-BrPA did not influence glucose kinase activity.

Although the bioenergetic profile indicates that cells with 3-BrPA have a low O_2_ consumption capacity, no significant changes are observed in mitochondrial membrane potential, as evidenced by Mitotracker staining, compared to control conditions (Figure 5). Nevertheless, glycosome staining reveals no difference between cells treated with or without 3-BrPA, indicating that glycosomes are not disrupted. In addition, the use of aldolase antibodies also demonstrated no commitment to aldolase expression despite glycolytic inhibition. These results suggest that 3-BrPA did not alter the structure of these organelles, although a significant functional disturbance was observed.

## 4. Discussion

*Trypanosoma cruzi* is a hemoflagellate and heteroxenous parasite that has a complex life cycle, transiting between vertebrate and invertebrate hosts, involving proliferative stages (amastigotes within mammalian cells and epimastigotes within the vector’s gastrointestinal tract) and non-proliferative stages (metacyclic trypomastigotes) [35]. Epimastigotes typically inhabit an environment where glucose is limited (the insect’s gut). However, they favor glucose over amino acids, even though L-proline and various other amino acids can be easily transported and utilized [36]. However, trypanosomatids exhibit a higher glucose consumption rate, which is linked to the peculiar characteristic of producing and releasing fermentative, still-reduced compounds from glucose breakdown into the surrounding medium, even in the presence of oxygen, rather than fully oxidizing glucose to carbon dioxide and water [37].

Cancer cells or protozoan cells require large quantities of energy to maintain elevated rates of proliferation, a characteristic of highly proliferative cells [17]. In this scenario, 3-BrPA is efficient once it targets the glycolytic pathway and mitochondrial function at micromolar concentrations [16,24,38]. There is no systemic toxicity in animals across a wide dosage range (5–25 mg/kg; [25]). Although 3-BrPA did not change epimastigotes’ proliferation or viability (Figure 1), as also demonstrated for amastigotes’ proliferation [22], a significant stimulus of ecto-phosphate activity is observed in cells maintained in the presence of 3-BrPA (Figure 2c). Inorganic phosphate (Pi) is an essential macronutrient for every living organism. It is not only essential for producing vital cellular components such as ATP, nucleic acids, phospholipids, and proteins, but it also participates in numerous metabolic processes, including energy transfer, protein synthesis, and carbon and amino acid metabolism [39]. Pi starvation may increase ecto-phosphatase expression in several microorganisms, including *T. cruzi*.

Additionally, elevated ecto-phosphatase activity may reflect alterations in energy metabolism, as one-time metabolic enzymes depend on Pi for their functionality [34,40]. However, the inhibition of Pi uptake (Figure 2a) or ecto-5’-nucleotidase (Figure 2d) suggests that, although the cells are energy-starved, they are unable to compensate for this starvation. Although this concentration did not affect cell viability, it had a substantial effect on energy metabolism, which, in association with other drugs, could be utilized as a potential therapeutic agent. Even in cancer cells, the association treatment is suggested, but not as the sole drug treatment choice. Metabolism-targeted drugs may prove particularly effective when used in combination with standard chemotherapy. A chemo-potentiating potential of 3-BrPA has been shown in combination with common anticancer drugs [41]. Moreover, the differences between life forms are important to highlight. Epimastigote forms are essentially proliferative and are found in a nutritionally poor environment in the insect vector. This form can utilize amino acids and fatty acids as sources of carbon and energy [42].

In *Toxoplasma gondii*, restricting the availability of Pi appears to cause a similar outcome, as knockout Pi transporter (ΔTgPiT) parasites increase the expression of bradyzoite markers, which include surface proteins and glycolytic enzymes. When the import of Pi is limited, ΔTgPiT parasites reduce the expression of succinyl coenzyme A synthase, an enzyme that converts GDP or ADP and Pi into GTP or ATP. In conditions of Pi scarcity, the downregulation of this enzyme could serve as a mechanism to reallocate intracellular Pi resources [28]. In the case of inefficient Pi uptake, low metabolism in the parasite can be observed.

Unlike other eukaryotic organisms, *T. cruzi* has a single mitochondrion that branches along the length of the cell body. Their “abnormal” metabolic design enables them to succeed in their hosts’ changing nutritional conditions [35]. Accordingly, cells maintained in the presence of 3-BrPA exhibit reduced O_2_ consumption compared to control cells (Figure 3a,b), reflecting the low intracellular ATP levels in these conditions (Figure 3c). Furthermore, reducing H_2_O_2_ production due to 3-BrPA reinforces the functional mitochondrial disruption effect (Figure 3d), showing that mitochondrial investigation is highly relevant due to the metabolic plasticity of *T. cruzi*. It is essential to note that, despite the cells exhibiting variations in mitochondrial functionality, they remain viable and display a proliferative profile compared to the control.

Moreover, intracellular ATP levels are reduced by 3-BrPA administration to the same extent as those due to hypoxia (Figure 3c), and there is no synergistic effect of 3-BrPA in hypoxic conditions. This data could indicate that 3-BrPA may promote hypoxia-like conditions. Hypoxia has already been demonstrated to lead to a proliferation stimulus while disrupting mitochondrial function [27]. Fermentative pathways could be stimulated [43].

*T. cruzi* presents glycosomes, organelles that contain not only enzymes of the glycolytic and gluconeogenic pathways but also several enzymes of other metabolic processes, including fermentation [44]. This compartmentalization is advantageous, since classical inhibitors do not regulate glycolytic enzymes, like glucokinase or phosphofructokinase-1, in trypanosomatids. Therefore, the glycosome maintains a high glycolytic flux [45]. It was observed that, although 3-BrPA inhibits GAPDH activity to the same level as iodoacetamide, a specific GAPDH inhibitor (Figure 4b), glucokinase activity is not modulated by 3-BrPA (Figure 4a), indicating that other pathways, such as the pentose phosphate pathway [46], may be utilizing glucose-6-phosphate. When the glycosome was observed using aldolase staining as a marker, no difference was observed between cells maintained with or without 3-BrPA (Figure 5). In glucose starvation, *T. cruzi* can switch from glycolytic-based metabolism to fatty acid oxidation. In this case, metabolic flexibility can maintain sufficient ATP synthesis for survival under stress [47]. Another alternative that can be used is glycerol as a carbon source during scarcity. In this case, uptaken glycerol can be phosphorylated by a glycosomal glycerol kinase, forming glycerol-3-phosphate (G3P). The latter can be used by mitochondrial glycerol-3-phosphate dehydrogenase or can be converted to DHAP by glycosomal glycerol-3-phosphate dehydrogenase [48]. Moreover, other carbohydrates, such as mannose, galactose, glucosamine, and xylose, can feed glycolysis at various points, ensuring the production of pyruvate and ATP [49]. Proline, for example, can be oxidized to pyruvate, supplying the mitochondrial oxidation [50]. Understanding this metabolic flexibility is crucial for developing better therapeutic approaches when targeting the metabolic energy pathway.

Furthermore, although 3-BrPA impaired O_2_ consumption as well as H_2_O_2_ and ATP production, no difference in mitochondrial staining was observed, indicating that the mitochondrial membrane potential was not affected. In *T. brucei*, ATP synthase can operate in a reverse mode, generating electrochemical membrane potential as it hydrolyzes ATP to pump protons across the membrane. In this mode, there is an inhibition of the oxidative phosphorylation pathway [51], which could explain why the mitochondrial membrane potential remains unchanged, while O_2_ consumption is affected.

In summary, it was observed that 3-BrPA inhibits mitochondrial function and GAPDH activity. Investigating mitochondrial disruption by 3-BrPA is highly relevant due to the metabolic plasticity of *T. cruzi*. However, cells grown in the presence of 3-BrPA have compensation in fermentation or other pathways, such as the pentose phosphate pathway (PPP), to sustain the energy required for normal epimastigote proliferation. Therefore, studies of 3-BrPA may lead to the elucidation of potential therapeutic applications, as it has an urgent need to seek new therapeutic targets for Chagas disease (Figure 6). These results imply the need to develop a combination therapy when treating Chagas disease, as *T. cruzi* can modulate its metabolism in response to therapeutic pressure, distinguishing it from other protozoan parasites. This adaptive metabolic behavior suggests that monotherapy targeting a single metabolic pathway may not be sufficient to eradicate the parasite. In this way, developing more effective and less toxic approaches that consider different disease mechanisms is important.

## Figures and Tables

**Figure 1 pathogens-14-00631-f001:**
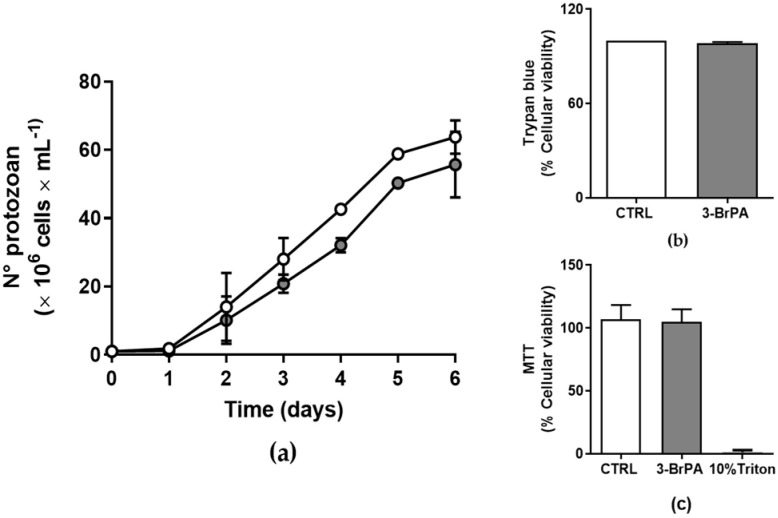
3-BrPA treatment did not affect the growth and viability of *T. cruzi* epimastigotes. (**a**) The presence (gray circles) or absence (open circles) of 3-BrPA in the LIT medium did not affect the growth of epimastigotes (*n* = 4). (**b**) 3-BrPA treatment did not alter parasite viability during the stationary phase (*n* = 3). (**c**) The MTT assay also confirmed cell viability. Treatment did not affect the viability of epimastigotes. A total of 1% Triton X-100 served as a negative control (*n* = 3).

**Figure 2 pathogens-14-00631-f002:**
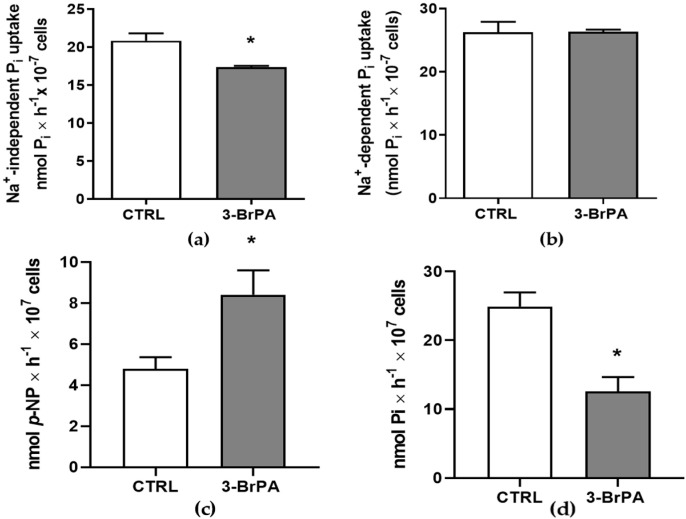
3-Bromopyruvate modulates membrane proteins in different ways. (**a**) Comparison between H^+^:Pi influx in epimastigotes grown with (gray bar) or without (white bar) 3-BrPA. Treated epimastigotes exhibited a reduction in H + Pi influx (*n* = 3). (**b**) Na^+^:Pi influx was reduced in epimastigotes grown with 3-BrPA (gray bar) when compared with control conditions (white bar) (*n* = 3). (**c**) 3-BrPA-treated epimastigotes displayed increased ecto-phosphatase activity (*n* = 5). (**d**) Ecto-5′-nucleotidase was also modulated by 3-BrPA administration; treated cells displayed lower ecto-5′-nucleotidase activity compared to the control group (*n* = 4) * *p*  <  0.05 compared with CTRL (unpaired Student’s *t*-test).

**Figure 3 pathogens-14-00631-f003:**
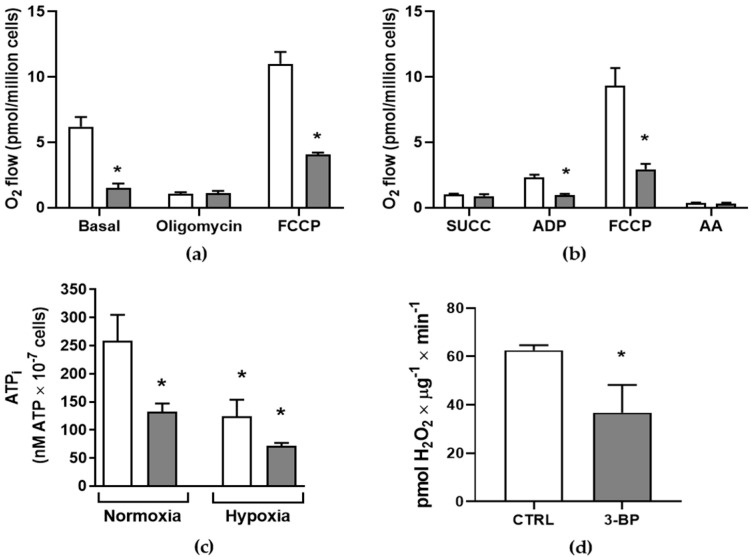
3-Bromopyruvate interferes in epimastigotes respiration. (**a**) Intact T. cruzi epimastigotes exposed to 3-BrPA (gray bar) displayed lower basal O_2_ consumption when compared with control conditions (white bars). Treatment also decreased maximum respiratory capacity (*n* = 3). (**b**) Epimastigotes were digitonin-permeabilized to measure mitochondrial activity. 3-BrPA abolishes the ADP stimulation in oxygen consumption and reduces the maximum respiratory capacity (*n* = 3). (**c**) 3-BrPA treatment reduced intracellular ATP levels when compared to control conditions (*n* = 4). (**d**) Reduced mitochondrial activity increases H_2_O_2_ production in treated parasites (*n* = 5). * *p*  <  0.05 compared with CTRL (one-way ANOVA followed by Tukey’s test).

**Figure 4 pathogens-14-00631-f004:**
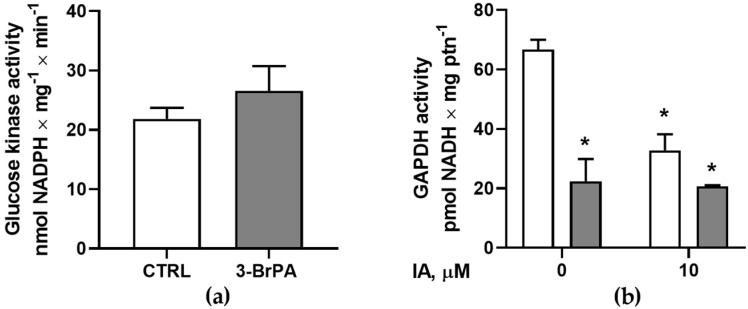
3-Bromopyruvate promotes glycolytic changes in epimastigotes. (**a**) Treated epimastigotes (gray bars) exhibited glucokinase activity values close to those of untreated cells (white bars) (*n* = 4). (**b**) 3-BrPA treatment results in the inhibition of GAPDH activity (gray bars) when compared with control conditions (white bars). Iodacetamide (IA, 10 µM) was used as an internal control in both conditions (*n* = 4). * *p* < 0.05 compared with CTRL (one-way ANOVA followed by Tukey’s test).

**Figure 5 pathogens-14-00631-f005:**
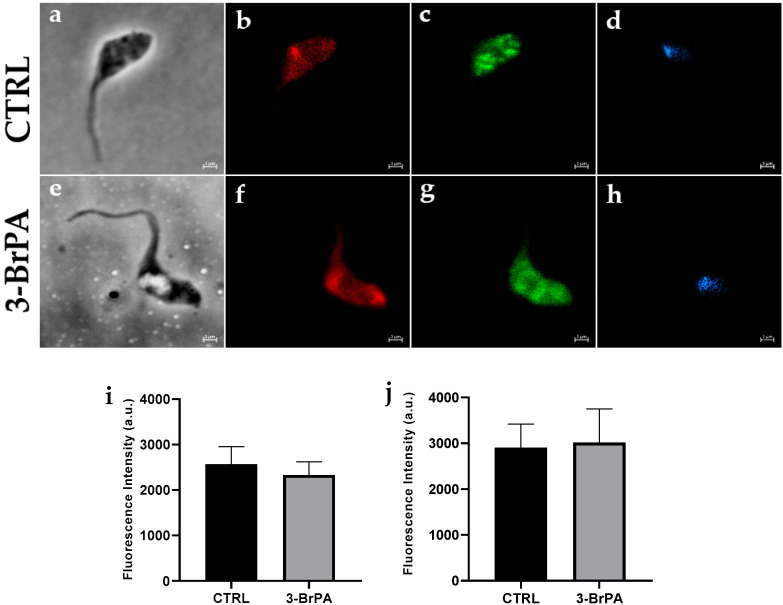
3-Bromopyruvate does not change mitochondrial potential or glycosomes. Immunofluorescence microscopy of *T. cruzi* epimastigotes grown in LIT medium with or without 300 µM 3-BrPA. Phase contrast (**a**,**e**). Mitochondrial membrane potential staining with MitoTracker ((**b**,**f**), red). Glycosome staining with anti-aldolase ((**c**,**g**), green). Nuclei staining with DAPI ((**d**,**h**), blue). Percentage of parasites with staining for Mitotracker (**i**) and aldolase antibody (**j**) in 90 parasites. Magnification 100×. Calibration bar 1 µm. Statistical analysis was performed with an unpaired Student *t*-test.

**Figure 6 pathogens-14-00631-f006:**
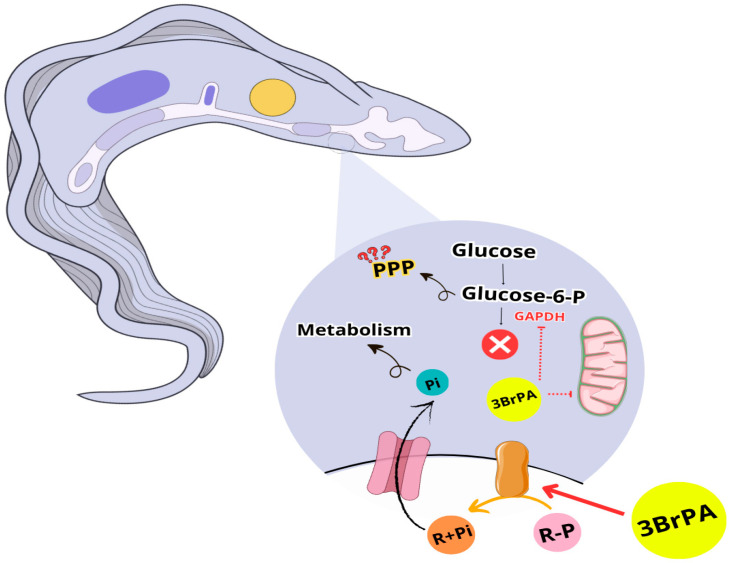
Proposed model for the effects of 3-Bromopyruvate on *Trypanosoma cruzi* metabolism. 3-BrPA stimulates ecto-phosphatese activity (brown protein), which hydrolyzes phosphorylated substrates (R-P), releasing Pi and the dephosphorylated product (R). Pi is taken up by a specific transporter (pink protein) and incorporated into metabolic pathways. At the intracellular level, 3-BrPA inhibits GAPDH, disrupting glycolysis and interfering with mitochondrial function. As a compensatory mechanism, the accumulation of glucose-6-phosphate may be redirected to other metabolic pathways, such as the pentose phosphate pathway (PPP), indicated by question marks. Red arrows: stimulation by 3-BrPA. Dashed red lines: inhibition by 3-BrPA. The figure was designed using Canva and icons from NIH Bioart.

## Data Availability

The original contributions presented in the study are included in the article. Further inquiries can be directed to the corresponding author.

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
