# Peer review of "3-Bromopyruvate Impairs Mitochondrial Function in *Trypanosoma cruzi"

_pathogens, 2025, doi:10.3390/pathogens14070631_

Round 1

Reviewer 1 Report

Comments and Suggestions for Authors

The current article “3-Bromopyruvate impairs mitochondrial function in Trypanosoma cruzi” by da Costa et al. presents an important topic in parasitology and drug development, namely the impact of 3-bromopyruvate, a known glycolytic inhibitor, on the energy metabolism and mitochondrial function of Trypanosoma cruzi. The research is timely and relevant, the experimental design is generally sound and includes diverse approaches. Although there are minor corrections and improvements.

I would suggest the following corrections:

1. Revise the language for grammar and structure.

2.Provide a more direct comparison and contrast of the results with those found in T. cruzi .

3. Expand on the discussion of potential compensatory metabolic pathways and their implications for therapy.

4. Moderate claims about clinical relevance.

5. Reduce self-citation below 10%. The IThenticate report also has 34%, try to reduce them.

6.Revise the discussion in order of potential toxicity, off-target effects, and feasibility of combination therapies.

Author Response

Reviewer #1:

General comment.

Citation. The current article “3-Bromopyruvate impairs mitochondrial function in Trypanosoma cruzi” by da Costa et al. presents an important topic in parasitology and drug development, namely the impact of 3-bromopyruvate, a known glycolytic inhibitor, on the energy metabolism and mitochondrial function of Trypanosoma cruzi. The research is timely and relevant, the experimental design is generally sound and includes diverse approaches. Although there are minor corrections and improvements. I would suggest the following corrections:

Answer: We thank the reviewer for the nice comment. We addressed the comments as follows.

Major comments.

Comment #1. Revise the language for grammar and structure.

Answer: As required, the English has been carefully revised and corrected in RM

Comment #2. Provide a more direct comparison and contrast of the results with those found in T. cruzi .

Answer: Thank you for this comment. We had better expand the discussion regarding the results observed and the literature in RM.

Comment #3. Expand on the discussion of potential compensatory metabolic pathways and their implications for therapy.

Answer: We thank the reviewer for this comment. As also requested by reviewer #3, we expand the discussion regarding compensatory metabolic pathways and their implications for therapy.

Comment #4. Moderate claims about clinical relevance.

Answer: Thank you for this comment. We now state that 3-BrPA is a potential candidate to be used as a trypanocide agent in association with other drugs.

Comment #5. Reduce self-citation below 10%. The IThenticate report also has 34%, try to reduce them.

Answer: We thank the reviewer for this comment. Most of the self-citation in OM was referred to methodology and could be easily replaced. Most references were removed or replaced appropriately in RM.

Comment #6. Revise the discussion in order of potential toxicity, off-target effects, and feasibility of combination therapies.

Answer: This discussion is fundamental to this work, and it was also asked for by reviewer 2. For this reason, in the discussion topic of RM, we clarify this issue.

Reviewer 2 Report

Comments and Suggestions for Authors

The present study reports the effect of 3-bromopyruvate (3Br-PA) on epimastigote forms of T. cruzi, mainly on energy metabolism.

In the natural life cycle of T. cruzi, the epimastigote (proliferative) forms are found in the insect vector, a nutritionally poor environment. Both blood-trypomastigotes (introduced from the hematophagy of the vertebrate host) and epimastigotes use amino acids and fatty acids as a source of carbon and energy (although, in in vitro culture, both forms preferentially use glucose, when available).

The rationale for carrying out the study is not clear from the introduction. According to the background on the subject, 3-BrPA shows antiproliferative activity in tumor cells, causing cell death. Evidence indicates that this substance inhibits the activity of enzymes of cellular energy metabolim such as hexokinase II, glyceraldehyde-3-phosphate dehydrogenase and succinate dehydrogenase (GAPDH).

Although in the abstract and title, the authors claim that 3-BrPA impairs mitochondrial function, the results presented in the study do not support this claim. The results using MitoTracker Red (used to evaluate the mitochondrial membrane potential) show that there was no difference between epimastigotes treated and not treated with the substance (lines 293-294). In addition, as stated by the authors, “The impairment of electron transport results in high reactive oxygen prodution”. (lines 254-255). However “3-BrPA induces low H2O2 production” (line 256 and Figure 3d (not 3c).

Nor is it possible to propose that 3-BrPA has a future application in Chagas disease based on the results presented.

Other issues that authors should consider:

-The life cycle of T. cruzi alternates between an invertebrate host (insect vector) and a vertebrate host (mammals, including man).

-The natural transmission route of T. cruzi is mediated by insect vectors of the Triatominae family. Although cases of transmission through contaminated food have been reported in recent literature, it is not the most common route.

-Unlike other eukaryotic organisms, T. cruzi has a single mitochondrion that branches along the cell body.

-Why was the concentration of 300 µM selected for the study? This concentration does not inhibit growth and does not interfere with the viability of epimastigotes, as shown in figure 1. Again, how could this substance be used to treat Chagas disease? Even with this result, the authors decided to invest in the substance's mechanism of action. If the substance interferes with mitochondrial function, why was there no difference in cell viability?

-“Modified LIT medium”. What is changed in the LIT culture medium?

-Positive controls for the reactions were not included.

-The references of the methodologies should be revised. For example, reference 31 refers to the protein quantification method (Lowry et al., 1951).

-There is no strong connection between the parameters analyzed and the results obtained in the discussion.

-The figure legends can be summarized. Some repeat the methodology several times.

-The English language must be revised.

Comments on the Quality of English Language

Proofreading the English language can improve the readability of the article.

Author Response

Reviewer #2:

General comments.

Citation. The present study reports the effect of 3-bromopyruvate (3Br-PA) on epimastigote forms of T. cruzi, mainly on energy metabolism. In the natural life cycle of T. cruzi, the epimastigote (proliferative) forms are found in the insect vector, a nutritionally poor environment. Both blood-trypomastigotes (introduced from the hematophagy of the vertebrate host) and epimastigotes use amino acids and fatty acids as a source of carbon and energy (although, in in vitro culture, both forms preferentially use glucose, when available).

Answer: We thank the reviewer for this comment. In RM, we notified this characteristic for the specific life forms.

Major comments.

Comment #1: The rationale for carrying out the study is not clear from the introduction. According to the background on the subject, 3-BrPA shows antiproliferative activity in tumor cells, causing cell death. Evidence indicates that this substance inhibits the activity of enzymes of cellular energy metabolim such as hexokinase II, glyceraldehyde-3-phosphate dehydrogenase and succinate dehydrogenase (GAPDH).

Answer: Not only was this inhibition observed in tumor cells but also in some protozoan cells, the so-called highly proliferative cells. Highly proliferative cells, such as protozoan parasites and cancer cells, rely on aerobic glycolysis to support the rapid generation of biomass, a phenomenon known as the Warburg effect in cancer cells. These eukaryotes that rapidly replicate utilize well-proven metabolic processes to effectively transform glucose and specific amino acids into biomass and energy at desired rates. That is why a potential antitumor drug could also be a potential trypanocide agent. This is now stated in the Introduction section on RM.

Comment #2: Although in the abstract and title, the authors claim that 3-BrPA impairs mitochondrial function, the results presented in the study do not support this claim. The results using MitoTracker Red (used to evaluate the mitochondrial membrane potential) show that there was no difference between epimastigotes treated and not treated with the substance (lines 293-294). In addition, as stated by the authors, “The impairment of electron transport results in high reactive oxygen prodution”. (lines 254-255). However “3-BrPA induces low H2O2 production” (line 256 and Figure 3d (not 3c).

Answer: This criticism is addressed in two parts:

  1. Mitotracker Red is used to measure mitochondrial membrane potential; however, mitochondrial functionality was also evaluated by oxygen consumption in intact (Figure 3b) or permeabilized (Figure 3a) cells. The results showed a significant difference in O2 consumption without changing ΔΨm. One explanation resides in the fact that some trypanosomatids, like T. brucei, can maintain ΔΨm without a functional oxidative phosphorylation pathway. For that, we added the sentence, “Furthermore, although 3-BrPA impaired O2 consumption and H2O2 and ATP production, no difference in mitochondrial staining was observed, indicating that the mitochondrial membrane potential was not affected. In T. brucei, ATP synthase can operate in a reverse mode, generating an electrochemical membrane potential as it hydrolyzes ATP to pump protons. In this mode, there is an inhibition of the oxidative phosphorylation pathway [52](Please check doi: doi: 10.1111/j.1432-1033.1992.tb17278.x.), which could explain why the mitochondrial membrane potential remains unchanged, while O2 consumption is affected.” in RM discussion topic, with the new reference #52.
  2. We revised the statement in OM, “The impairment of electron transport results in high reactive oxygen production,” to “The impairment of electron transport could interfere with reactive oxygen production,” in RM to clarify this issue.

Comment #3: Nor is it possible to propose that 3-BrPA has a future application in Chagas disease based on the results presented.

Answer: Thank you for this comment. We now state that 3-BrPA is a potential candidate to be used as a trypanocide agent in association with other drugs.

Comment #4: Other issues that authors should consider:

-The life cycle of T. cruzi alternates between an invertebrate host (insect vector) and a vertebrate host (mammals, including man).

Answer: The life cycle was better detailed in the Introduction section of RM.

Comment #5: -The natural transmission route of T. cruzi is mediated by insect vectors of the Triatominae family. Although cases of transmission through contaminated food have been reported in recent literature, it is not the most common route.

Answer: The oral transmission route is the most frequent in some endemic regions, such as Brazil, which is a significant concern for Public Health, as stated by governmental websites.  We clarify this information in RM with the sentence: “In Brazil, approximately 76% of CD cases were transmitted orally in 2021, while the vector transmitted 7%.”

Comment #6: -Unlike other eukaryotic organisms, T. cruzi has a single mitochondrion that branches along the cell body.

Answer: We added this information to the discussion topic, highlighting the importance of this unique metabolism.

Comment #7: -Why was the concentration of 300 µM selected for the study? This concentration does not inhibit growth and does not interfere with the viability of epimastigotes, as shown in figure 1. Again, how could this substance be used to treat Chagas disease? Even with this result, the authors decided to invest in the substance's mechanism of action. If the substance interferes with mitochondrial function, why was there no difference in cell viability?

Answer: We address this comment in two parts:

  1. The concentration of 300 µM was used based on literature. Cancer cells or protozoan cells require large quantities of energy to maintain elevated rates of proliferation, a characteristic of highly proliferative cells (Please check doi: 10.3390/membranes13010042). In this scenario, 3-BrPA is efficient once it targets the glycolytic pathway and mitochondrial function at micromolar concentrations (Please check doi: 10.1042/BJ20080805; doi: 10.1016/j.exppara.2021.108154; doi: 10.1016/j.bbrc.2004.09.047). There is no systemic toxicity in animals across a wide dosage range (5–25 mg/kg; please refer to doi: 10.1007/s10637-008-9145-0). A new reference (25) has been added to the reference management (RM).
  2. Although this concentration did not affect cell viability, it had a substantial effect on energy metabolism, which, in association with other drugs, could be utilized as a potential therapeutic agent. Even in cancer cells, the association treatment is suggested, not as the sole drug treatment choice. Metabolism-targeted drugs are particularly effective when used in combination with standard chemotherapy. A chemo potentiation potential of 3-BP has been demonstrated in combination with common anticancer drugs (please see doi: 10.1007/s10863-012-9419-2). We better discuss this issue on discussion topic in RM.

Comment #8: -“Modified LIT medium”. What is changed in the LIT culture medium?

Answer: We thank the reviewer for this comment. We removed the word “modified”; it was used in the common LIT culture medium.

Comment #9: -Positive controls for the reactions were not included.

Answer: We thank the reviewer for this comment. Most of the methodology used in this work is based on standard curves as references. We added this missing information to the M&M section on RM. In the case of positive or negative controls, this information is also stated in the new Materials and Methods (M&M) section.

Comment #10: -The references of the methodologies should be revised. For example, reference 31 refers to the protein quantification method (Lowry et al., 1951).

Answer: The references were double-checked after the notification by reviewers.

Comment #11: -There is no strong connection between the parameters analyzed and the results obtained in the discussion.

Answer: We now expand on the discussion topic to address this criticism.

Comment #12: -The figure legends can be summarized. Some repeat the methodology several times.

Answer: The figure legends were summarized after the criticism.

Comment #13: -The English language must be revised.

Answer: As required, the English has been carefully revised and corrected in RM

Reviewer 3 Report

Comments and Suggestions for Authors

This manuscript investigates the impact of 3-bromopyruvate (3-BrPA) on mitochondrial function and energy metabolism in Trypanosoma cruzi, the causative agent of Chagas disease. The authors show that 3-BrPA impairs mitochondrial oxygen consumption, reduces intracellular ATP, and inhibits GAPDH activity, while epimastigote proliferation remains unaffected. The findings suggest the parasite employs compensatory metabolic adaptations, making 3-BrPA a potential candidate for combinatorial therapies.The study addresses a crucial area in antiparasitic therapy by exploring drug repurposing for a neglected tropical disease. Investigating mitochondrial disruption by 3-BrPA is highly relevant due to the metabolic plasticity of T. cruzi.

Major comments

1) While the methodology is generally thorough, some key elements (e.g., number of replicates for all experiments, precise buffer compositions for some assays, and image quantification strategies) require more detail to ensure reproducibility. For instance, the statistical treatment of microscopy data could be improved.

2) data quantification from immunofluorescence microscopy (Figure 5) would enhance rigor.

Minor comments

  1. Lines 153–162: The ATP quantification methodology is clear but lacks information on internal controls and normalization strategies. Were values normalized per total protein or per cell number?
  2. Line 285–289 (Figure 4): Despite noting GAPDH inhibition, the mechanistic implications of this inhibition (i.e., downstream metabolite flux) are only superficially addressed in the discussion.
  3. The manuscript is mostly well-written, though some sentences would benefit from minor grammar or syntax corrections for improved clarity. For example, rephrasing "the 3-BrPA induces a lower ecto-nucleases activity" to "3-BrPA reduces ecto-nuclease activity" would be smoother.
  4. Minor typographical and formatting inconsistencies (e.g., inconsistent use of Greek symbols like µM) should be corrected during proofreading.

Author Response

Reviewer #3:

General comments.

Citation. This manuscript investigates the impact of 3-bromopyruvate (3-BrPA) on mitochondrial function and energy metabolism in Trypanosoma cruzi, the causative agent of Chagas disease. The authors show that 3-BrPA impairs mitochondrial oxygen consumption, reduces intracellular ATP, and inhibits GAPDH activity, while epimastigote proliferation remains unaffected. The findings suggest the parasite employs compensatory metabolic adaptations, making 3-BrPA a potential candidate for combinatorial therapies.The study addresses a crucial area in antiparasitic therapy by exploring drug repurposing for a neglected tropical disease. Investigating mitochondrial disruption by 3-BrPA is highly relevant due to the metabolic plasticity of T. cruzi.

Answer: We thank the reviewer for the nice comment.

Major comments.

Comment #1) While the methodology is generally thorough, some key elements (e.g., number of replicates for all experiments, precise buffer compositions for some assays, and image quantification strategies) require more detail to ensure reproducibility. For instance, the statistical treatment of microscopy data could be improved.

Answer: Thank you for this comment. In figure legends of RM, the number of replicates for all experiments is now stated. In M&M of RM, we also clarify the buffer composition. Moreover, for microscopy analysis, we now provide image quantification.

Comment #2) data quantification from immunofluorescence microscopy (Figure 5) would enhance rigor.

Answer: As suggested, we added the new Figures 5i and 5j with immunofluorescence quantification.

Minor comments.

Comment #3) Lines 153–162: The ATP quantification methodology is clear but lacks information on internal controls and normalization strategies. Were values normalized per total protein or per cell number?

Answer: Thank you for your comment. We have now added this information to the M&M section.

Comment #4) Line 285–289 (Figure 4): Despite noting GAPDH inhibition, the mechanistic implications of this inhibition (i.e., downstream metabolite flux) are only superficially addressed in the discussion.

Answer: We thank the reviewer for this comment. As also requested by reviewer #1, we expand the discussion regarding compensatory metabolic pathways and their implications for therapy.

Comment #5) The manuscript is mostly well-written, though some sentences would benefit from minor grammar or syntax corrections for improved clarity. For example, rephrasing "the 3-BrPA induces a lower ecto-nucleases activity" to "3-BrPA reduces ecto-nuclease activity" would be smoother.

Answer: Thank you for your comment. The manuscript was carefully revised, and we improved clarity in the Results and Methods sections.

Comment #6) Minor typographical and formatting inconsistencies (e.g., inconsistent use of Greek symbols like µM) should be corrected during proofreading.

Answer: Typography and formatting were double-checked in R1.

Round 2

Reviewer 1 Report

Comments and Suggestions for Authors

The authors addressed all my comments for this paper. The paper has undergone significant improvement after revision.

Reviewer 2 Report

Comments and Suggestions for Authors

Dear authors, thank you for all your replies.

The changes made to the manuscript have improved its readability and comprehension.

The experimental controls were included in the tests, and the presentation of the results is clearer and more objective. The discussion now shows coherence and consistently articulates all the findings of the study.

I have a few more comments:

-Lines 78-80: the sentence needs to be improved. The way it is written, there are still doubts as to why the study was carried out.

“However, it was not demonstrated that the efficacy of 3-BrPA against T. cruzi proliferation of intracellular amastigote forms [23], and the effects on metabolism were not evaluated.”

-2.1 I suggest using verbs in the past tense

-Minor comments:

Please standardize use: intra-macrophages or intramacrophages (lines 72-73)

Please add italic to T. brucei

Author Response

Reviewer #2:

General comment.

Citation. Dear authors, thank you for all your replies.

The changes made to the manuscript have improved its readability and comprehension.

The experimental controls were included in the tests, and the presentation of the results is clearer and more objective. The discussion now shows coherence and consistently articulates all the findings of the study.

I have a few more comments:

Major comments.

Comment #1.-Lines 78-80: the sentence needs to be improved. The way it is written, there are still doubts as to why the study was carried out.

“However, it was not demonstrated that the efficacy of 3-BrPA against T. cruzi proliferation of intracellular amastigote forms [23], and the effects on metabolism were not evaluated.”

Answer: We thank the reviewer for this comment. We now modified the sentence for: “Although 3-BrPA shows no effects on T. cruzi intracellular amastigote forms proliferation [23], possible impacts on parasite metabolism were not evaluated.”

Comment #2.-2.1 I suggest using verbs in the past tense

Answer: The alteration was done as suggested.

Minor comments.

Comment #3. Please standardize use: intra-macrophages or intramacrophages (lines 72-73)

Answer: In R2, it is now used only intramacrophages.

Comment #4. Please add italic to T. brucei

Answer: We have double-checked all the scientific names, and they are now in italics.

Reviewer 3 Report

Comments and Suggestions for Authors

I have no further comments